# Non-asymptotic transients away from steady states determine cellular responsiveness to dynamic spatial-temporal signals

**Akhilesh Nandan, Aneta Koseska** [ID] *

Cellular computations and learning, Max Planck Institute for Neurobiology of Behavior – caesar, Bonn, Germany

* aneta.koseska@mpinb.mpg.de

## Abstract

Majority of the theory on cell polarization and the understanding of cellular sensing and responsiveness to localized chemical cues has been based on the idea that non-polarized and polarized cell states can be represented by stable asymptotic switching between them. The existing model classes that describe the dynamics of signaling networks underlying polarization are formulated within the framework of autonomous systems. However these models do not simultaneously capture both, robust maintenance of polarized state longer than the signal duration, and retained responsiveness to signals with complex spatial-temporal distribution. Based on recent experimental evidence for criticality organization of biochemical networks, we challenge the current concepts and demonstrate that non-asymptotic signaling dynamics arising at criticality uniquely ensures optimal responsiveness to changing chemoattractant fields. We provide a framework to characterize non-asymptotic dynamics of system's state trajectories through a non-autonomous treatment of the system, further emphasizing the importance of (long) transient dynamics, as well as the necessity to change the mathematical formalism when describing biological systems that operate in changing environments.

**Data Availability Statement:** All codes are available on Github: https://github.com/akhileshpnn/SubPB-mechanism.

## Author summary

During wound healing or embryonic development, cells in tissues or organs migrate over large distances by sensing local chemical cues. The migration response is based on cell polarization—the formation of a distinct front and back of the cell in the direction of the chemical cues. These cues are however disrupted and have a complex spatial-temporal profile. This suggests that cell polarity must be robustly established in signal direction, but also flexibly adapt when signals change. A large diversity of abstract and biochemically detailed models have been proposed to explain cell polarity, but they cannot fully describe the experimental observations. Here, we argue that cell polarization is a highly dynamic transient process, and must be studied via an explicit time-dependent form. We demonstrate that criticality organization uniquely enables formation of metastable polarized states that can be robustly maintained for a transient period even when the signals are

**Funding:** This work was supported by the Max Planck Society through the Lise Meitner excellence program awarded to AK, which supported AK and AN. The funders had no role in study design, data collection and analysis, decision to publish, or preparation of the manuscript.

**Competing interests:** The authors have declared that no competing interests exist.

disrupted, but also enable rapid adaptation to temporal or spatial signal changes. Using a combination of bifurcation and quasi-potential landscape analysis, we provide a framework to characterize non-asymptotic transients explicitly, and thereby further emphasize the necessity to change the mathematical formalism when describing biological systems that operate in changing environments.

## Introduction

During embryogenesis, wound healing, or cancer metastasis, cells continuously sense and chemotactically respond to dynamic spatial-temporal signals from their environment [1–5]. This response is based on cell polarization—the formation of a distinct front and back of the cell through stabilization of polarized signaling activity at the plasma membrane. Broad range of cells, including epithelial or nerve cells, fibroblasts, neutrophils, *Dictyostelium discodeum* etc., display multiple common polarization features: quick and robust polarization in the direction of the localized signal, sensing of steep and shallow gradients (and subsequent amplification of the internal signaling state between the opposite ends of the cell), as well as threshold activation as a means to filter out noise [6, 7]. Moreover, polarity and thereby directional migration is transiently maintained after the trigger stimulus is removed (memory in polarization), but the cells remain sensitive to new stimuli, and can rapidly reorient when the signal's localization is changed. In response to multiple stimuli such as two sources with varying concentrations, rapid resolution with a unique axis of polarity towards the signal with higher concentration is ensured [7].

A large diversity of models, both abstract and biochemically detailed have been proposed, that however cannot fully describe the experimental observations. For example, the local excitation global inhibition model (LEGI) [8, 9] and its variants [10–12] rely on an incoherent feed-forward motif, whose dynamics doesn't account for transient memory in polarization. The Turing-like models based on a local activation long-range inhibition (activator-inhibitor system) [13, 14] are not robust to noise, cannot resolve simultaneous signals in physiologically relevant time frame [6], or maintain responsiveness to upcoming signals with same or different spatial localization. The Wave-pinning model on the other hand is based on a higher-order nonlinear positive feedback [15–18], and in contrast to the Turing-like models, can account for cell re-polarization (polarity reversal) upon change of signal localization. The robustness of the re-polarization is however conditioned on the signal strength and width [19]. However, it has not been studied whether the Wave-pinning model allows to integrate signals that do not change in space but are disrupted over time, as expected during a cell migration in complex tissue environment. To address in particular cell responsiveness to disrupted and/or signals with complex temporal and spatial distribution, we have recently proposed a mechanism, referred to as a SubPB mechanism, that relies on critical organization to a saddle-node which stabilizes a subcritical pitchfork (*PB*) bifurcation ($SN_{PB}$) [20]. We have demonstrated also experimentally using the Epidermal growth factor receptor (EGFR) network, that the SubPB mechanism enables navigation in complex environments due to the presence of metastable "ghost" of the polarized state, which gives the system both a memory of previous signals, but also flexibility to respond to signal changes.

We take here the conceptual basis of polarity one step further: we argue that cell polarization and responsiveness necessary for navigation in changing spatial-temporal chemoattractant fields is a highly dynamic transient process, and must be studied via an explicit time-dependent form, or as a non-autonomous process. For non-autonomous systems, both the

number and the position of steady states change, implying that the steady-state behavior alone does not fully capture the dynamics of the system. What is most relevant are therefore the trajectories representing the change of the state of the system that follow the steady-state landscape changes. This conceptual shift enables to consider transients explicitly, and we demonstrate that a pure non-autonomous succession of steady states, as characteristic for the LEGI, Turing-like or Wave-pinning models cannot explain both transient memory in cell polarization and cellular responsiveness to upcoming signals. On the other hand, non-asymptotic transient states that arise due to organization at criticality, as in the SubPB mechanism enable to maintain the dynamics of the sensing network away from a fixed point, and uniquely confer optimal sensing and responsiveness to cells that operate in a changing environment. We therefore argue that the formal descriptions how cells sense and respond to dynamic signals must be modified to consider also (long) non-asymptotic transient processes.

## Results

### Studying cell polarity response as transients in non-autonomous systems

To investigate the dynamical characteristics of polarization, we consider a generalized reaction-diffusion (RD) system in a one-dimensional domain with two components $u$ and $v$,

$$\frac{\partial u(\theta, t)}{\partial t} = f_u(u, v) + D_u \frac{\partial^2 u}{\partial \theta^2} + s(\theta, t)v$$

$$\frac{\partial v(\theta, t)}{\partial t} = f_v(u, v) + D_v \frac{\partial^2 v}{\partial \theta^2} - s(\theta, t)v$$

(1)

where $(\theta, t) \in R$ are angular position on the plasma membrane of a cell with respect to its center and time, $f_u, f_v: R \times R \to R$ are the reaction terms of $u$ and $v$ respectively, $D_u$ and $D_v$—the diffusion constants, and $s(\theta, t)$—the distribution of the external chemoattractant signal with respect to the cell. The reaction term $f_u(u, v)$ is chosen as for the Wave-pinning model [15], exemplifying a Rho-GTPase cycle with an inter conversion between its active, membrane bound ($u$) and inactive, cytosolic ($v$) components (Fig 1A, top):

$$f_u(u, v) = (k_0 + \gamma u^2/(K^2 + u^2))v - \delta u$$

(2)

and $f_v = -f_u$ due to mass conservation $\int_0^L (u + v)\, dl = Lc_{total}$, with $L = 2\pi R \mu m$—the total length of the one-dimensional domain of the cell perimeter, $R$—cell radius. The positive feedback from $u$ onto its own production (via GEFs) is represented by a Hill function of order 2 with maximal conversion rate $\gamma$ and saturation parameter $K$, $k_0$ is a basal GEF conversion rate and $\delta$ is the constant inactivation rate (via GAPs). This model exhibits a subcritical pitchfork bifurcation [16].

To analyze the dynamical features of the system from aspect not only of the bifurcation structure, but also the quasi-potential landscapes as a means to characterize the system's transitions in the presence of complex spatial-temporal signals, we have simplified the cell geometry by considering a one-dimensional projection consisted of two bins (left, right) such that $u_L$, $u_R$, $v_L$, $v_R$ can be exchanged, mimicking species' diffusion (Fig 1A bottom). When subjected to an analytical treatment, the resulting ODE system Eq (3) demonstrates the existence of a subcritical pitchfork bifurcation, equivalently to the full RD model ([16], Methods). Additionally, numerical bifurcation analysis [21] of the system Eq (3) in absence of a signal shows that the subcritical $PB$ is stabilized via $SN_{PB}s$ at a critical total concentration of the system's constituents ($c_{total}^{critical}$, Fig 1B). The $PB$ generates a transition from a non-polar or homogeneous steady state (HSS, $u_{L,s} = u_{R,s}$, $v_{L,s} = v_{R,s}$), to a polar state represented as a inhomogeneous steady state

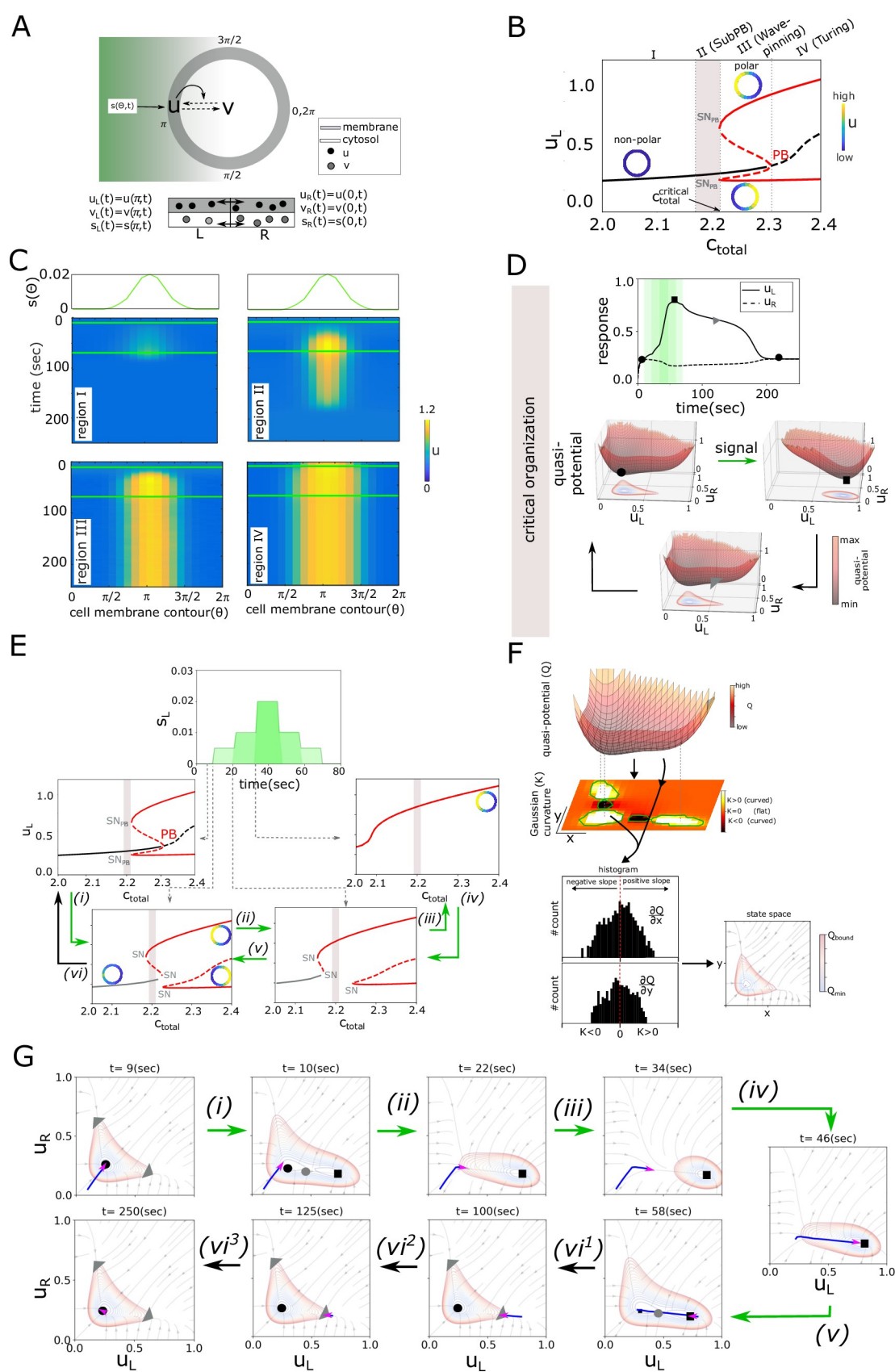

**Fig 1. Describing polarization response using transients in a non-autonomous system.** (A) Top: Schematic representation of the two component ($u(\theta, t)$—membrane bound, $v(\theta, t)$—cytosolic) reaction-diffusion system Eq (1) of a circular cell stimulated with spatial chemoattractant gradient $s(\theta, t)$. ($0-2\pi$): angular positions with respect to cell center. Solid/dashed arrows: causal/conversion link. Bottom: respective one dimensional projection of the model Eq (3), with left (L) and right (R) bin. Double-headed arrows: diffusion-like exchange. (B) Bifurcation diagram of the system in (A), Eq (3), with respect to the total protein concentration $c_{total}$, in absence of a signal. Dotted lines: regions I-IV with qualitatively different dynamical response; solid /dashed lines: stable/unstable homogeneous (black) and inhomogeneous (red) steady states. Insets: schematic representations of the homogeneous (non-polar)/inhomogeneous (polar) states. *PB*: Pitchfork bifurcation, $SN_{PB}$: Saddle-node of a pitchfork bifurcation. Shaded region: criticality/SubPB mechanism. Parameters: $k_0 = 0.067s^{-1}$, $\gamma = 1s^{-1}$, $K = 1\mu M$, $\delta = 1s^{-1}$, $\tilde{D}_u = 0.01sec^{-1}$, $\tilde{D}_v = 10sec^{-1}$. (C) Spatial chemoattractant distributions (top) and the corresponding kymograph of $u(\theta, t)$, obtained using the RD model Eq (1), for organization in regions I-IV in (B). Green horizontal lines: gradient duration. For all RD simulations, the width at half maximum of $s(\theta, t)$ is set to 25% of the cell perimeter (unless specified). $c_{total} = 2.1; 2.21; 2.26; 2.32\mu M$ for regions I-IV respectively, $D_u = 0.1\mu m^2/sec$, $D_v = 10\mu m^2/sec$, and other parameters as in (B). (D) Top: Time series of $u$ (corresponding to region II in (B)). Green shaded region: signal interval. Circle/square/triangle: non-polar/polar/transient-polar (memory) state. Bottom: Respective quasi-potential landscape transitions. Coloured contour maps: landscape projections in $u_L$-$u_R$ plane. Green/black arrows:transitions during signal presence/absence. (E) Unfolding of the *PB* in the presence of a spatial step-like signal ($s_R = 0.0$). The bifurcation diagram for each step-wise increment is shown. Green/black arrows as in (D). Gray line: marginally asymmetric steady state, line descriptions as in (B). (F) Representation of Gaussian curvature estimate of a quasi-potential landscape (top), schematic representation of the subsequent slope distributions for distinct landscape regions and resulting contours determining the phase space region characterised with asymptotic behavior of the system's trajectory (bottom). See also S1(D) Fig. (G) Corresponding instantaneous phase portraits with the integrated progression of the trajectory (blue line) during the last 40s before signal change. Pink arrow: current trajectory position and direction. Black/grey circles: HSS/saddle; squares—IHSS; triangles—dynamical "ghost" / memory state. For (D-G), equations and parameters as in (B).

(IHSS). The IHSS is manifested via two symmetric branches—a combination of a high $u$ at the cell left and low at the right side of the cell ($u_R < u_L$ or left-polarized, top branch), and $u_R > u_L$ or right-polarized (lower branch). Thus, depending on $c_{total}$, 4 distinct organization regimes are possible (I-IV in Fig 1B).

To study how the system responds to transient gradient stimulus for organization in the different regimes, we calculated the kymographs representing the spatial-temporal $u$ distributions using the RD simulations (Eq (1)), the signal $s(\theta, t)$ is introduced as a Gaussian distribution along the circular membrane (S1(A) Fig, Methods). Moreover, we also tracked the changes in the system's dynamics by estimating the quasi-potential energy landscape changes [22, 23] using the one-dimensional projection model (Eq (3)) with a signal introduced as a step-like function with amplitudes ($s_L$, $s_R$) (S1(B) Fig, Methods). When the system is organized in region I, a transient gradient stimulus does not lead to robust polarization (kymograph in Fig 1C). The quasi-potential landscapes demonstrate that increasing the signal amplitude only shifts the position of the stable homogeneous steady state (the geometry of the landscape changes, S1 (C) Fig, top). This leads to a marginal local increase in $u$, without breaking the system's symmetry. In region III on the other hand, a transient gradient signal irreversibly shifts the system to the stable polarized state (Fig 1C). Formally, this regime corresponds to the previously described Wave-pinning model [15–17]. In this region, both the non-polar (homogeneous) and the polarized (inhomogeneous) steady states coexist (Fig 1B). Thus in absence of a signal, the quasi-potential landscape is characterized by three minima—one corresponding to the HSS (circle), and the other two corresponding to the IHSS branches (left- and right-polarized states, S1(C) Fig, middle). Upon signal addition, the minimum corresponding to the HSS disappears, leaving a one-well quasi-potential landscape of the stable polar state. Signal removal reverts the system to the the three well quasi-potential landscape, however the system remains in the IHSS well, leading to a sustained polarization. Similar steady state transitions are also observed for organization in region IV, however the systems here starts from a pre-polarized state, as only the IHSS is stable (Fig 1C and S1(C) Fig, bottom). Formally, this regime corresponds to a Turing-like mechanism of polarization [16]. Thus for organization in regions III and IV, the system doesn't reset to basal non-polar state after a transient stimulus.

In contrast, when the system is organized at criticality (before $SN_{PB}$, shaded region II in Fig 1B, what we refer to as the SubPB mechanism), a transient gradient stimulus leads to rapid $u$ polarization in the direction of the maximal chemoattractant concentration. The polarized state is only transiently maintained after signal removal, corresponding to a temporal memory of the signal direction (Fig 1C and 1D (top)). The changes in the quasi-potential landscape further clarify these tranisitions: in absence of a signal, only the HSS (the non-polar state, single well) is stable (Fig 1D, HSS: circle). However, since the system is positioned close to the critical transition towards the IHSS, the landscape also has an area with a shallow slope. Upon signal addition, the topology of the landscape changes. The HSS is lost and the IHSS (Fig 1D, square) is stabilized, causing the transition to the newly established well. The opposite transition then takes place upon signal removal, but in this case, the system is initially transiently trapped in the region with the shallow slope (Fig 1D, triangle), which is manifested as a transient memory of the polarized state. This transient trapping dynamically occurs from a "ghost" of a saddle-node bifurcation which is lost when the signal is removed [20, 24].

These observed change of the topology of the system's phase space suggests that cell polarization should be formally treated as a non-autonomous process. In general, in non-autonomous systems, either the geometry (change in the positioning, shape and size of the attractors), or the topology (change in the number or stability of the attractors) of the underlying phase space is altered [22]. To gain deeper insight in the quasi-potential landscape changes in the presence of a transient signal, we calculated next the bifurcation diagrams during the subsequent increase/decrease in the signal amplitude. Even a low-amplitude spatial signal (step *(i)*) introduces an asymmetry to the system and thereby a universal unfolding of the *PB* [25], such that a marginally asymmetric steady state (Fig 1E, gray solid lines) replaces the HSS (black solid lines in signal absence). Moreover, for the same parameter values, now also the IHSS (a remnant of the *PB* that disappeared) is also stable. Increasing the spatial signal's amplitude in the next steps leads to an increase in the extent of the unfolding, rendering the IHSS as the only stable solution at the maximal signal strength (step *(iii)* in Fig 1E). This solution corresponds to the single-well landscape in Fig 1D, representing a robust polarization of the system. Decreasing the signal amplitude in the same step-wise manner results in the reversed changes in the bifurcation diagram structure, thereby explaining the resetting the system to the non-polar HSS after signal removal.

The non-autonomous treatment of the systems thus allowed us to track the changes in the number and stability of the attractors, however the fixed point analysis does not capture the full dynamics of the system, as the memory emerging from the $SN_{PB}$ 'ghost' cannot be examined in this analysis. This implies that the transient dynamics of the system must be considered explicitly, through the trajectories of the system which represent the change of the state of the system. To classify the nature of the transients, it is necessary to quantify the phase space regions in which the steady states asymptotically bind the trajectories. For this, we calculated the Gaussian curvature of the quasi-potential landscapes for each step of the signal (schematic in Fig 1F, top). A surface has a positive curvature ($K > 0$) at a point if the surface curves away from that point in the same direction relative to the tangent to the surface, a negative ($K < 0$) —if the surface curves away from the tangent plane in two different directions, and a $K \sim 0$— a flat surface. Complementing the curvature calculations with the slopes along each point (e.g. for a well, positive curvature and slope values distributed around 0 are a unique identifier, Fig 1F, bottom and S1(D) Fig) allowed to identify the phase space region where the trajectory asymptotically moves towards the steady state (contour plot in Fig 1F, bottom). Thus, movement of the system's trajectory in areas outside of these regions correspond to a non-asymptotic, transient dynamics of the system. Detailed phase plane analysis of the system for each signal amplitude showed that during transition *(i)*, the marginally asymmetric steady state and

the IHSS are stable (as shown in as in Fig 1E), whereas the system's trajectory is trapped in the former one (Fig 1G). In the next steps *(ii, iii)*, only the IHSS attractor is stable but moves its position. The trajectory's current state falls behind and reacts by travelling towards the moving attractor. Since the flow rate along the trajectory is smaller than the velocity of the attractor movement, the system is not able to catch up with the moving steady state and temporally reverts from asymptotic to transient behaviour. The trajectory is asymptotically bounded to the IHSS only at the highest signal strength (step *(iv)*). Decreasing the signal strength leads to re-appearance of the marginally asymmetric state, whereas the IHSS moves from the previous step, such that the trajectory reverts the direction to follow the attractor (step *(v)*). At a zero signal amplitude, the topology of the landscape changes again such that a single stable HSS is generated. In the position where the IHSS attractor was lost however, the landscape is characterized with a shallow slope ("ghost" of the $SN_{PB}$, triangle in Fig 1G). This lies right outside the border determining the asymptotic behavior, and the system's trajectory not only lags behind, but it is effectively trapped in this state for a transient period of time (($vi^1$, $vi^2$)) resulting in transient memory of the polarized state, before it reverts to the HSS attractor (($vi^3$)). Thus, examining transient dynamics during the signal-induced transition reveals important details that the shape of the trajectory, and hence the response of the system, could not be understood by focusing only on the steady state behaviour.

## Non-autonomous succession of steady states underlies the dynamics of the existing polarity models

We next examined the dynamical mechanisms underlying the LEGI, Turing-like and the Wave-pining cell polarity models. The bifurcation analysis was performed using the linear perturbation analysis LPA, [26, 27], which allows to identify the dynamical transitions in RD models characterized with large disparity between the diffusivity of the system's components (see Methods for details). As can be already deduced by the LEGI network topology—the incoherent feed-forward motif, this model (Eq (18)) has a single HSS (Fig 2A, shaded region: respective parameter organization as used in the literature [8, 10]). The Turing-like model (Eqs (16) and (20)) on the other hand displays a transcritical bifurcation (*TC*) at a critical total concentration of the system's constituents. The *TC* marks a transition from non-polar HSS to a polarized, symmetry-broken state. In the literature [13, 14], the model is parameterized after the *TC*, where the HSS is unstable (Fig 2B, shaded region: parameter organization). Such organization makes the Turing model dynamically equivalent to organization in region IV in Fig 1B. The Wave-pinning model, as described in [15], corresponds to organization in the region where the HSS and the IHSS co-exist (Fig 2C, shaded region; equivalent to region III in Fig 1B). RD simulations of these models, consistent with previous findings, demonstrate that upon transient gradient stimulation, the LEGI model shows a transient polarization that decays to homogeneous non-polar state immediately after stimulus removal. In contrast, both the Turing-like and the Wave-pinning models showed a long-term maintenance of the polarized state after signal removal (S2(A) Fig).

Considering the time-dependence explicitly in the analysis shows that the trajectory describing the state of the LEGI model exposed to transient spatial signal asymptotically follows the change in the position of the only steady state of the system, thereby marking the steady-state as the only relevant behavior (Fig 2D and 2E and S2(B) Fig). As noted by the bifurcation analysis, the Turing-like model is organized in the stable symmetry-broken state, thereby cannot describe a stable non-polar state. Thus, the non-autonomous analysis in this case is equivalent to that of the SubPB model for organization in region IV (S1(C) Fig). Due to the organization in the region where the HSS and the IHSS coexist on the other hand, the

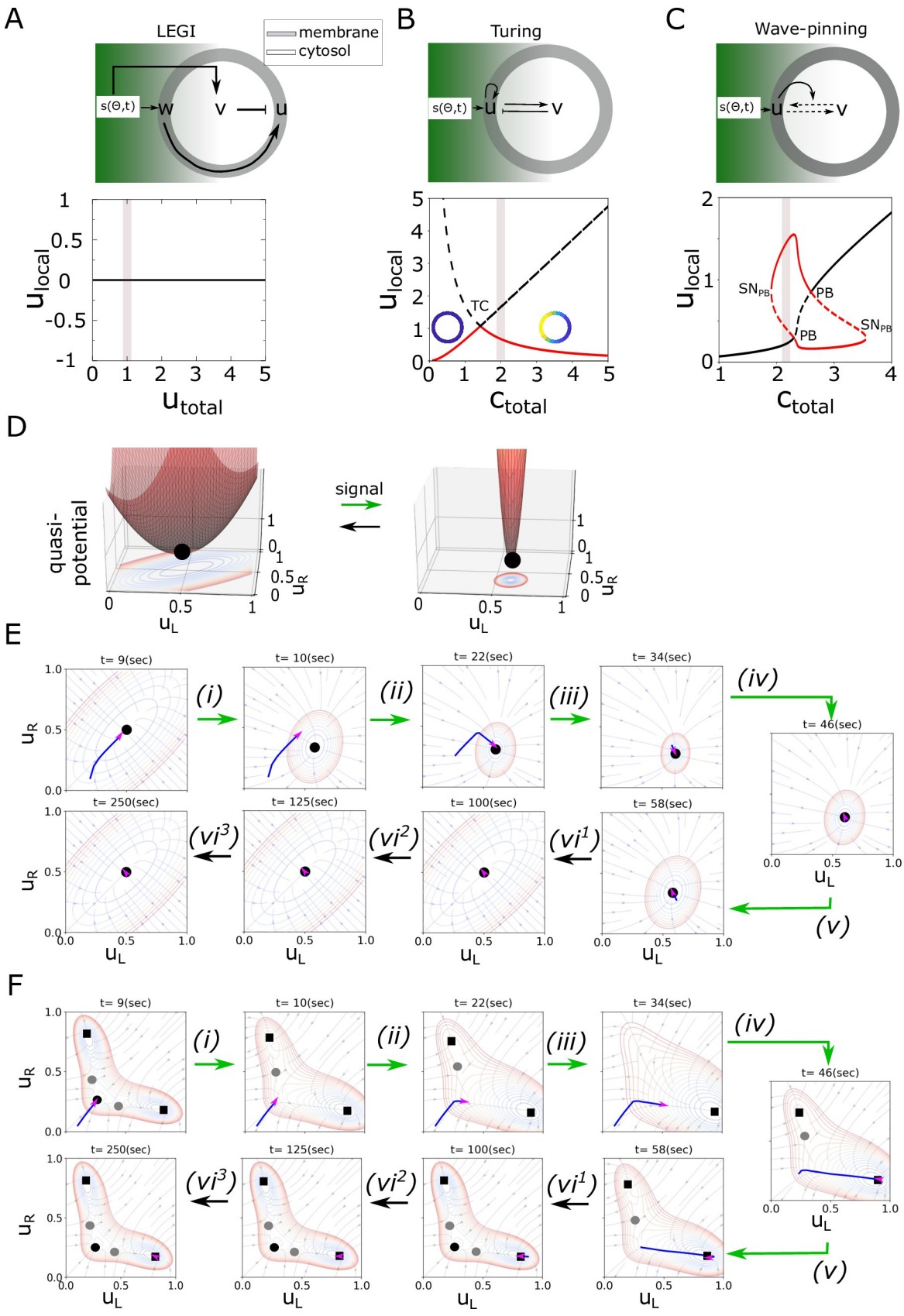

**Fig 2. Dynamical characteristics of the LEGI, Turing-like and Wave-pining polarity models.** (A) Top: topology of interaction of LEGI model. Color coding and arrows as in Fig 1A. Bottom: corresponding bifurcation diagram and the respective parameter organization in signal absence. Simulations have been performed using Eq 18, and $k_u = k_{-u} = 2s^{-1}$, $k_v = k_{-v} = 1s^{-1}$, $k_w = k_{-w} = 1\mu M^{-1}s^{-1}$. (B), (C) same as in (A) but for Turing and Wave-pinning models, respectively. The simulations of the Turing model correspond to Eqs (16) and (20), with $a_1 = 2.5$, $a_2 = 0.7$, and for the Wave-pinning model, Eqs (2) and (16) and parameters as in Fig 1B. In (A)-(C), shaded region: parameter organization, *TC*: Transcritical bifurcation, *PB*: pitchfork bifurcation, $SN_{PB}$: saddle-node; *u/w* are membrane bound, and *v*—cytosolic component, $u_{local}$: local variable associated with *u* from LPA analysis, line description as in Fig 1B. (D) Quasi-potential landscapes calculated for the LEGI model (Eq 19) subjected to a transient signal. Landscapes in absence and maximal signal strength are shown. Transitions in signal presence/absence: green/black arrows. Coloured contour maps: landscape projection in $u_L$-$u_R$ plane. (E) Corresponding instantaneous phase portraits and system's trajectory (as in Fig 1G). Black circles: stable steady states. Transitions in signal presence/absence: green/black arrows. (F) Same as in (E) only for the Wave-pinning model. Grey circles: saddles; black squares: IHSS.

Wave-pinning model can explain both, the non-polar and the polar state [15]. Non-autonomous analysis of the Wave-pinning model however demonstrates that it is fully characterized by an asymptotic behavior, realized through non-autonomous switching between the available steady states (Fig 2F). Thus, the LEGI, Turing-like and Wave-pinning models are characterized by a qualitatively different dynamics in comparison to the SubPB model: asymptotic behavior towards the available steady states in contrast to the non-asymptotic, transient dynamics complemented with transient trapping by the dynamical "ghost", which temporarily maintains the system away from the steady state.

## Responsiveness to spatial-temporal signals is optimally enabled by the transient dynamics and metastable state in the SubPB model

To investigate the difference in basic polarization features for the different models, we quantified next from the RD simulations a polarization ratio ($\frac{u_{\theta=\pi}}{u_{\theta=0}}$) to steep and shallow gradients which are quantified via a stimulus difference, $sd = (s_{\theta = \pi} - s_{\theta = 0}) \times 100$; time to reach stable polarization at a threshold signal amplitude that induces polarization, and polarization ratio in response to signals with an increasing offset. Scaling of the models to reflect physiological time-scales was implemented as in [6].

The RD simulations showed that for the LEGI and Turing-like models, polarization can be induced even when the gradient steepness is < 0.5% between the front and the back of the cell (Fig 3A). However, the polarization ratio achieved by the LEGI-type model is relatively small ($\approx 1$), indicating that the LEGI mechanism cannot account for signaling amplification when sensing shallow gradients. This is a direct consequence of the underlying dynamical mechanism: an external signal triggers a continuous and reversible re-positioning of the only stable attractor, and therefore cannot account for signaling amplification (Fig 2A, 2D and 2E). The Turing-type model also showed polarization for very low stimulus differences, which results from organization after the *TC*, region where the non-polar state is unstable. The Wave-pinning model on the other hand effectively generated robust polarization response. However, the response could be triggered even for low gradient amplitudes. This can be explained again by the dynamical structure: due to the organization where HSS and IHSS coexist, a "hard" signal-induced transition effectively results in a threshold activation ($sd_{thresh} = 0.3\%$). That the Turing and the Wave-pinning models could be activated at low stimulus difference across the cell suggests that these models are also susceptible to spurious activation. This could be further demonstrated in the presence of fluctuations around the homogeneous steady state (mimicking noisy initial conditions, S3(A) Fig, Methods). Thus, these models do not exhibit reliable threshold activation and are thereby not robust to noise. In terms of the polarization times on the other hand, the LEGI- and Turing-type models

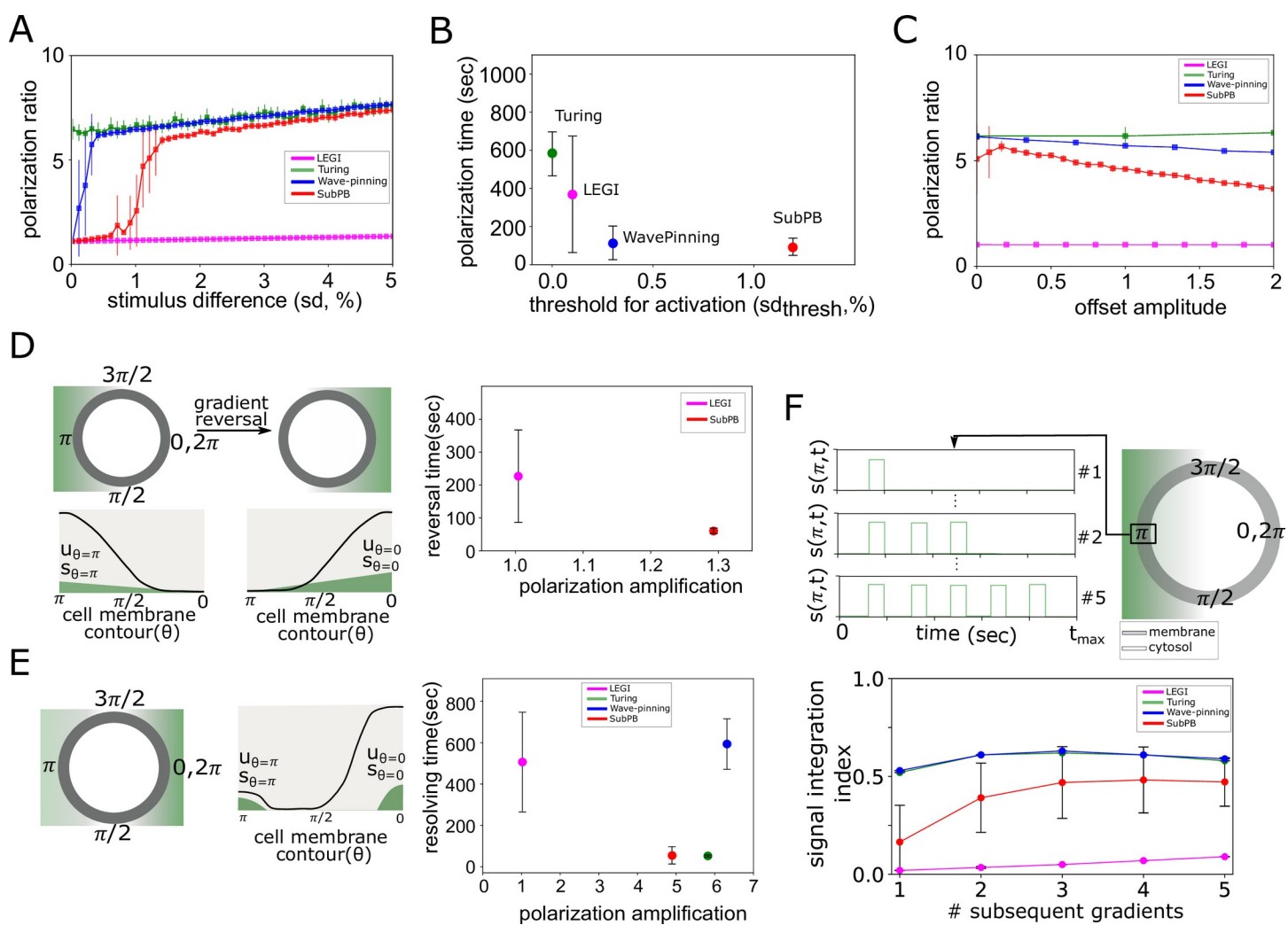

**Fig 3. SubPB mechanism enables optimal responsiveness to spatial-temporal chemoattractant signals.** (A) Average polarization ratio ($\frac{u_{\theta=\pi}}{u_{\theta=0}}$) as a function of a stimulus difference across the cell (sd = ($s_{\theta=\pi} - s_{\theta=0}$) × 100)). Mean ± s.d from 10 RD simulation repetitions. (B) Minimal threshold that activates the system ($sd_{thresh}$) and time to achieve stable polarization (Methods). (C) Polarization ratios upon stimulation with a gradient with an offset (S3(B) Fig). For (B-C), colours as in (A). (D) Left: schematic representation of gradient reversal across the cell, and respective representation of the spatial profiles of $u(\theta, t)$ and $s(\theta, t)$. Right: Quantification of polarization reversal time and the respective polarization amplification ($\frac{u_{\theta=0}}{u_{\theta=\pi}}$) upon stimulation with $\frac{s_{\theta=0}}{s_{\theta=\pi}} = 2$. LEGI and Turing-like models did not demonstrate re-polarization in time-interval of 1000 sec. (E) Left: Schematic representation for numerical stimulation protocol with simultaneous signals localized on opposite ends, and corresponding schematic spatial profile of $u(\theta, t)$ and $s(\theta, t)$. Right: Quantification of the time necessary to achieve unique polarization axis and the corresponding polarization amplification, for stimulus ratio $\frac{s_{\theta=0}}{s_{\theta=\pi}} = 2$. (F) Top: Schematic representation for numerical stimulation protocol with consecutive transient gradient stimuli from same direction. Bottom: Corresponding quantification of signal integration index (see Methods). See also S3 Fig.

polarized on a time scale longer than 6min, whereas the Wave-pinning model displayed rapid polarization (< 3min, Fig 3B). Moreover, testing the polarization responses to gradients with different offset demonstrated that, with exception of the LEGI, the remaining models robustly polarized under these conditions (Fig 3C and S3(B) Fig). The equivalent quantifications for the SubPB model on the other hand show that it responds to steep and relatively shallow gradients, threshold activation and thereby robustness to noisy signal activation ($sd_{thresh}$ = 1.2%), rapid polarization times (<3min), and robust polarization to

gradients with offset (Fig 3A–3C and S3(A) and S3(B) Fig). That the SubPB model displays optimal polarization features can be explained with the criticality organization: in absence of a signal, the non-polar state is the only stable steady state, thus threshold activation can be robustly achieved, whereas the subcritical nature of the *PB* gives rise to the signal amplification to shallow signals. Taken together, these results demonstrate that SubPB enables optimal polarization response.

We next tested the re-polarization capabilities of each of the models by subjecting the systems in the RD simulations to a spatial gradient until stable polarization was achieved, after which the gradient direction was reversed and its maximal amplitude was set to $2 \times sd_{thresh}$. The Turing- and Wave-pinning-type models did not re-polarize (in physiologically relevant time-frame, Fig 3D and S3(C) Fig). This can be understood from the non-autonomous analysis of the system (Fig 2F): the trajectory remained trapped in the symmetry-broken state after signal removal, such that rapid re-polarization cannot be achieved. The LEGI model re-polarized in a time-frame >3min, but the polarization ratio did not depend on the signal amplitude. The SubPB on the other hand, not only enabled rapid re-polarization to spatially reversed gradient signals (<1 min, Fig 3D and S3(C) Fig), but the polarization response was also sensitive to the amplitude of the reversed signal as reflected in the polarization amplification. In contrast to the Wave-pinning model, re-polarization for the SubPB mechanism is possible due to organization at criticality—after signal removal, the system is maintained in the dynamical "ghost" state (in contrast to the stable IHSS for the Wave-pinning model), thus the system can rapidly respond to the reversed signal and thereby quickly re-polarize. Additional analysis on the Wave-pinning and the SubPB model by systematically scanning the stimulus width and maximal amplitude of the re-polarization gradient showed that re-polarization in the Wave-pinning model is possible only for signals $\approx 7 \times sd_{thresh}$. In contrast, the re-polarization in the SubPB mechanism can be achieved for reversed gradients with wide range of widths and amplitudes (S3(D) Fig). When presented with two simultaneous, but distinctly localized signals with different amplitudes, only the SubPB and Wave-pinning mechanisms demonstrated effective resolving and rapid and robust polarization in the direction of the stronger signal (Fig 3E and S3(E) Fig). In contrast, both LEGI-, Turing- models required more than four times longer time to resolve the signals and polarize in direction of the stronger signal.

We next tested how the models respond to consecutive transient gradient stimulation from same direction, mimicking signals that are disrupted. This reflects the capability of the models to integrate signals with complex temporal profile, and adapt the duration of the polarized state accordingly. The response in the LEGI model rapidly decayed after signal removal, demonstrating complete absence of memory. Thus the system responds to each gradient independently (S3(F) Fig), as also reflected in the low signal integration index (Fig 3F), Methods. The Turing and the Wave-pinning models maintain the polarized state on a long-term after signal removal, thus they are insensitive to consecutive gradient signal stimulation from same direction: stimulation with a single or multiple consecutive signals does not change the total duration in which the polarized state is maintained, resulting in a constant signal integration index. In contrast, the SubPB model displays signal-integration features, adapting the polarization duration depending on the number of consecutive stimuli. These results therefore demonstrate that the SubPB mechanism uniquely enables sensing and responsiveness to dynamic signals, as a result of the critical organization that allows utilizing transient dynamics via the presence of a dynamical "ghost" state to adapt to dynamic signals in the environment.

## Discussion

We have demonstrated that it is necessary to consider transient dynamics and explicit time-dependence in order to describe cellular responsiveness to spatial-temporal chemoattractant signals. The current models in the literature rely on an autonomous system's description, where the system's topology determines the number, stability and type of available steady-states, whereas the external signals are thought only to induce switching between them. Description of the asymptotic behavior at or near a steady state is also attractive from mathematics point of view, as it provides a tractable analysis of the system using linear stability analysis [28]. However, this view only accounts for robustness of regulatory processes, ignoring the temporal system's changes. As we have shown here, the steady-state view cannot account for cellular responsiveness to dynamic cues or how cells resolve simultaneous signals, crucial features of cells that operate in the dynamic environments of tissues and organs.

In contrast, one of the basic characteristics of non-autonomous systems is that the quasi-potential landscape is dynamic itself under time-varying signals, resulting in changes in the number and stability of the steady states. These landscape changes thus guide the movement of the system's trajectory. For system's organization at criticality, as demonstrated here, a non-asymptotic transient behavior emerges upon the landscape changes, enabling the system to maintain both robustness (i.e. by transient trapping or slow motion in specific landscape region), while maintaining flexibility in the responses to upcoming cues. Indeed, recent experimental evidence has demonstrated that cell's protein activity dynamics is maintained away from steady state, thereby enabling them to retain transient memory of the previous signal's localization, while being responsive to newly perceived signals [20].

However, a general theory to analyze or formally describe non-asymptotic transient dynamics is lacking, and current analysis has been mostly limited to systems with regular external forcing [29], or numerical investigation of simple two dimensional models [22]. Here, we provide an additional tool (although also applicable mainly to low-dimensional systems) based on combination of extended bifurcation analysis and quantification of the Gaussian curvature of the landscape and the corresponding point-wise slope distribution, to separate quantitatively asymptotic from non-asymptotic behavior. Importantly, this framework enables to identify manifolds with a specific topology that maintain the system for a prolonged period of time away from the steady state. Such long non-asymptotic transients have been characterized in neuronal networks and have been particularly informative, not only about the identity and temporal features of the external signals [30], but also about basic forms of learning such as signal associations [31]. Stable heteroclinic channels have been proposed as an underlying dynamical mechanism that generates long stable transients in neuronal models [32]. Moreover, transient phenomena with much longer time scales have been also described in the context of regime shifts due to anthropogenic global changes in ecological systems [33]. We hereby argue that it is necessary to shift the description of biochemical computations in single cells towards non-autonomous system's description and focus on the role of transient dynamics for processing and interpreting spatial-temporal varying signals.

## Methods

### Analytical treatment of the SubPB model

Let us consider the system Eq (1) with reaction terms as in Eq (2). This describes the Wave-pinning model [15], and the SubPB model discussed here. To identify analytically the existence of a sub-critical *PB*, as well as to further calculate the quasi-potential landscapes, we consider a simplified one-dimensional projection where the cell constitutes of two bins (left, right)

between which the species can be exchanged:

$$\frac{du_L}{dt} = G_1(u_L, v_L, u_R) = \quad f_u(u_L, v_L) - \tilde{D}_u(u_L - u_R)$$

$$\frac{dv_L}{dt} = G_2(u_L, v_L, v_R) = \quad f_v(u_L, v_L) - \tilde{D}_v(v_L - v_R)$$

$$\frac{du_R}{dt} = G_3(u_L, u_R, v_R) = \quad f_u(u_R, v_R) - \tilde{D}_u(u_R - u_L)$$

$$\frac{dv_R}{dt} = G_4(v_L, u_R, v_R) = \quad f_v(u_R, v_R) - \tilde{D}_v(v_R - v_L)$$

(3)

The subscripts $L$ and $R$ stand for the two bins (Fig 1A, bottom), $\tilde{D}_u$ and $\tilde{D}_v$ are the diffusion-like terms, and $G_1$—$G_4$ combine the reaction-diffusion terms. Let $\mathbf{U_s} = \left[u_{L,s}, v_{L,s}, u_{R,s}, v_{R,s}\right]^T$ be the stable homogeneous steady (non-polar) state of the system ($u_{L,s} = u_{R,s}, v_{L,s} = v_{R,s}$). Stability of this state can be probed using a linear perturbation of the form $\mathbf{U}(t) = \mathbf{U_s} + \delta\mathbf{U}(t)$, where $\delta\mathbf{U} = \left[\delta u_L, \delta v_L, \delta u_R, \delta v_R\right]^T \exp(\lambda t)$, is a small amplitude perturbation with growth rate $\lambda$. Plugging this into Eq (3) gives the linearized equation:

$$\lambda \begin{bmatrix} \delta u_L \\ \delta v_L \\ \delta u_R \\ \delta v_R \end{bmatrix} \exp(\lambda t) = \quad \mathbf{J}' \begin{bmatrix} \delta u_L \\ \delta v_L \\ \delta u_R \\ \delta v_R \end{bmatrix} \exp(\lambda t)$$

(4)

where $\mathbf{J}'$ is evaluated at $\mathbf{U_s}$, and is given by:

$$\mathbf{J}' = \begin{bmatrix} \dfrac{\partial G_1}{\partial u_L} & \dfrac{\partial G_1}{\partial v_L} & \dfrac{\partial G_1}{\partial u_R} & 0 \\[2mm] \dfrac{\partial G_2}{\partial u_L} & \dfrac{\partial G_2}{\partial v_L} & 0 & \dfrac{\partial G_2}{\partial v_R} \\[2mm] \dfrac{\partial G_3}{\partial u_L} & 0 & \dfrac{\partial G_3}{\partial u_R} & \dfrac{\partial G_3}{\partial v_R} \\[2mm] 0 & \dfrac{\partial G_4}{\partial v_L} & \dfrac{\partial G_4}{\partial u_R} & \dfrac{\partial G_4}{\partial v_R} \end{bmatrix}$$

(5)

The occurrence of zero-crossing eigenvalues leads to either pitchfork or saddle-node bifurcations, and the solution for $\lambda = 0$ can be readily obtained by taking the well-defined limit $\lambda \to 0$ [34]. The existence of the PB bifurcation is related to the odd mode of the perturbation ($\delta u_L = -\delta u_R$ and $\delta v_L = -\delta v_R$) due to the symmetry of this bifurcation. Substituting these constrains in Eq (4) gives:

$$0 = \quad \mathbf{J}' \begin{bmatrix} \delta u_L \\ \delta v_L \\ -\delta u_L \\ -\delta v_L \end{bmatrix}$$

(6)

The symmetry in the perturbation further reduces the dimensionality of the Eq (6):

$$0 = \mathbf{F}_\lambda \begin{bmatrix} \delta u_L \\ \delta v_L \end{bmatrix} \tag{7}$$

where

$$\mathbf{F}_\lambda = \begin{bmatrix} \left(\dfrac{\partial G_1}{\partial u_L} + \dfrac{\partial G_3}{\partial u_R}\right) - \left(\dfrac{\partial G_1}{\partial u_R} + \dfrac{\partial G_3}{\partial u_L}\right) & \left(\dfrac{\partial G_1}{\partial v_L} + \dfrac{\partial G_2}{\partial v_R}\right) \\ \left(\dfrac{\partial G_2}{\partial u_L} + \dfrac{\partial G_4}{\partial u_R}\right) & \left(\dfrac{\partial G_2}{\partial v_L} + \dfrac{\partial G_4}{\partial v_R}\right) - \left(\dfrac{\partial G_2}{\partial v_R} + \dfrac{\partial G_4}{\partial v_L}\right) \end{bmatrix} \tag{8}$$

The linear system in Eq (7) has non-trivial solution only if the determinant of $\mathbf{F}_\lambda = 0$:

$$|\mathbf{F}_\lambda| = \begin{vmatrix} \left(\dfrac{\partial G_1}{\partial u_L} + \dfrac{\partial G_3}{\partial u_R}\right) \partial\left(\dfrac{\partial G_1}{\partial u_R} + \dfrac{\partial G_3}{\partial u_L}\right) & \left(\dfrac{\partial G_1}{\partial v_L} + \dfrac{\partial G_2}{\partial v_R}\right) \\ \left(\dfrac{\partial G_2}{\partial u_L} + \dfrac{\partial G_4}{\partial u_R}\right) & \left(\dfrac{\partial G_2}{\partial v_L} + \dfrac{\partial G_4}{\partial v_R}\right) - \left(\dfrac{\partial G_2}{\partial v_R} + \dfrac{\partial G_4}{\partial v_L}\right) \end{vmatrix} = 0 \tag{9}$$

where $|.|$ denotes the determinant of the matrix. The parameter values of Eq (3) that satisfies the condition in Eq (9) corresponds to the symmetry breaking $PB$.

To identify next whether the $PB$ is of sub-critical type, and thereby identify the presence of a $SN_{PB}$, a weakly nonlinear analysis of system Eq (1) must be performed to obtain a description of the amplitude dynamics of the inhomogeneous state. This can be achieved using an approximate analytical description of the perturbation dynamics based on the Galerkin method [35–37]. For simplicity, we outline the steps for a reaction-diffusion system in a one-dimensional domain. As we are interested in the description of a structure of finite spatial size (i.e. finite wavelength $k$ of the symmetry-broken state), the final solution of the system Eq (1) is expanded around the fastest growing mode, $k_m$ into a superposition of spatially periodic waves:

$$u(\theta, t) = \phi(t)e^{ik_m\theta} + \phi^*(t)e^{-ik_m\theta} + u_0(t) + \sum_{n=2}^{3}(u_n(t)e^{nik_m\theta} + u_n^*(t)e^{-nik_m\theta})$$

$$v(\theta, t) = \phi(t)e^{ik_m\theta} + \phi^*(t)e^{-ik_m\theta} + v_0(t) + \sum_{n=2}^{3}(v_n(t)e^{nik_m\theta} + v_n^*(t)e^{-nik_m\theta}) \tag{10}$$

where $u(v)_n(t)$ is the complex amplitude of the $n^{th}$ harmonics. The expansion is taken to $n = 3^{rd}$ order, rendering an amplitude equation of $5^{th}$ order. For simplification, the Hill function in $f_u(u, v)$ is approximated by assuming $(K/u) >> 1$ to yield $f_u(u, v) = (k_0' + \gamma' u^2)v - \delta u$ where $k_0' = \frac{k_0}{K^2}$ and $\gamma' = \frac{\gamma}{K^2}$. Substituting Eq (10) in Eq (1) gives,

$$\frac{d\phi}{dt}e^{ik_m\theta} + \frac{du_0}{dt} + .. = k_0'(\phi e^{ik_m\theta} + v_0..) + \gamma'((3|\phi|^2\phi + 2u_0v_0\phi)e^{ik_m\theta} +$$

$$+2(u_0 + v_0)|\phi|^2 + \ldots) - \delta(\phi e^{ik_m\theta} + u_0 + ..) - D_u(k_m^2\phi e^{ik_m\theta} + v_0 + ..) \tag{11}$$

Collecting coefficients of harmonics up to first order on either side gives an equation that governs the evolution of the amplitude:

$$\frac{d\phi}{dt} = (k'_0 - (D_u k_m^2 + \delta))\phi + 3\gamma'|\phi|^2\phi + 2\gamma' u_0 v_0 \phi$$

$$\frac{du_0}{dt} = (2\gamma'|\phi|^2 - \delta)u_0 + (k'_0 + 2\gamma'|\phi|^2)v_0$$

(12)

The complex coefficients of the $n = 0^{th}$ harmonics is next approximated as power series of $\phi(t)$ [35]:

$$u_0(t) \approx u_0^{(2)}|\phi|^2 + \ldots$$

$$v_0(t) \approx v_0^{(2)}|\phi|^2 + \ldots$$

(13)

Eq (13) is then substituted into Eq (12) giving:

$$\frac{d\phi}{dt} = (k'_0 - (D_u k_m^2 + \delta))\phi + 3\gamma'|\phi|^2\phi + 2\gamma' u_0^{(2)} v_0^{(2)} |\phi|^4 \phi$$

$$\frac{du_0}{dt} = (2\gamma'|\phi|^2 - \delta)u_0^{(2)}|\phi|^2 + (k'_0 + 2\gamma'|\phi|^2)v_0^{(2)}|\phi|^2$$

(14)

Higher order amplitudes were assumed to be in quasi-steady state, thus $\frac{du_0}{dt} = 0$, rendering $v_0^{(2)} \propto -u_0^{(2)}$. Substituting this into Eq (14) yields an approximated expression for $\phi$:

$$\frac{d\phi}{dt} = c_1\phi + c_2\phi^3 - c_3\phi^5$$

(15)

where $c_1 = (k'_0 - (D_u k_m^2 + \delta))$, $c_2 = 3\gamma'$ and $c_3 = 2\gamma'(u_0^{(2)})^2$. Eq (15) is of Stuart-Landau type and represents a normal form of a sub-critical pitchfork bifurcation. Taken together, this guarantees the existence of $SN_{PB}$ for system Eq (3).

## Local perturbation analysis (LPA)

Local perturbation analysis is a method to identify dynamical transitions in spatially-extended system [26, 27]. The method can be applied to any system where the two species $((u, v))$ are characterized with at least order-of-magnitude difference between their diffusivity, i.e $D_v >> D_u$. In such a case, it is possible to consider the limit $D_u \to 0$, $D_v \to \infty$, further allowing to probe the stability of the HSS of the PDE system under study (Eq (1) for $s(\theta, t) = 0$) with respect to a local perturbation in the form of a narrow peak of the slow variable with a negligible total mass. Thus, the height of this peak can be represented as a local variable ($u_{local}(t)$) that does not spatially spread. Due to the fast rate of diffusion of $v$, it can be represented by a uniform global quantity $v_{global}(t)$. Since $u$ does not spread and $v$ is uniform on the domain, $u$ can then be represented on the remainder of the domain (away from the perturbation) by a global quantity, $u_{global}(t)$, which for mass-conservation systems as in Eq (1) also captures the

evolution of $v_{global}(t)$:

$$\frac{\mathrm{d}u_{local}}{\mathrm{d}t} = f_u(u_{local}, (c_{total} - u_{global}))$$

$$\frac{\mathrm{d}u_{global}}{\mathrm{d}t} = f_u(u_{global}, (c_{total} - u_{global})) \tag{16}$$

Such systems can be further analyzed by means of classical (numerical) bifurcation analysis.

## Description of the different cell polarity models

**LEGI model.** The LEGI-type model system is characterized by an incoherent feed forward loop topology, where $w$ is the membrane bound activator, $v$ is the cytosolic inhibitor and $u$ is the membrane bound response component [8]. The equations are given by,

$$\frac{\partial w(\theta, t)}{\partial t} = f_w(w) + k_w s(\theta, t) + D_w \frac{\partial^2 w}{\partial \theta^2}$$

$$\frac{\partial v(\theta, t)}{\partial t} = f_v(v) + k_v s(\theta, t) + D_v \frac{\partial^2 v}{\partial \theta^2} \tag{17}$$

$$\frac{\partial u(\theta, t)}{\partial t} = f_u(w, u, v) + D_u \frac{\partial^2 u}{\partial \theta^2}$$

with $f_w(w) = -k_{-w}w, f_v(v) = -k_{-v}v, f_u(w, u, v) = k_u w(u_{total} - u) - k_{-u}vu$, and $s(\theta, t)$ is the external stimulus.

Applying LPA on this system, we obtain:

$$\frac{\mathrm{d}w_{local}}{\mathrm{d}t} = f_w(w_{local}); \frac{\mathrm{d}w_{global}}{\mathrm{d}t} = f_w(w_{global})$$

$$\frac{\mathrm{d}v_{global}}{\mathrm{d}t} = f_v(v_{global})$$

$$\frac{\mathrm{d}u_{local}}{\mathrm{d}t} = f_u(w_{local}, u_{local}, v_{global}) \tag{18}$$

$$\frac{\mathrm{d}u_{global}}{\mathrm{d}t} = f_u(w_{global}, u_{global}, v_{global})$$

The one dimensional projection of LEGI model is given by,

$$\frac{du_L(t)}{dt} = f_u(w_L^{qss}, u_L, v^{qss}) - \tilde{D}_u(u_L - u_R)$$

$$\frac{du_R(t)}{dt} = f_u(w_R^{qss}, u_R, v^{qss}) - \tilde{D}_u(u_R - u_L) \tag{19}$$

with

$$v^{qss} = 0.5 \frac{k_v}{k_{-v}} (s_L + s_R)$$

$$w_R^{qss} = \frac{k_w}{k_{-w}} (s_L + s_R) - w_L^{qss}$$

$$w_L^{qss} = \frac{k_w}{(2\tilde{D}_w + k_{-w})}\left(s_L + \frac{\tilde{D}_w(s_L + s_R)}{k_{-w}}\right)$$

This two component simplification was obtained from Eq (17) after a quasi-steady state approximation of $v$ and $w$.

**Turing model.** For the Turing-like model, the reaction term was taken from [13]:

$$f_u(u, v) = a_1\left(v - \frac{(u + v)}{(a_2(u + v) + 1)^2}\right) \tag{20}$$

with $f_v = -f_u$ (mass conservation). The external signal $s(\theta, t)$ was introduced same as in Eq (1), in contrast to [13], where $s(\theta, t)$ was introduced in the denominator of the reaction term.

## Estimating quasi-potential landscapes

In order to obtain the quasi-potential landscapes for the systems Eqs (3) and (19), the method described in [23] is adopted. For non-equillibrium systems, the underlying potential that defines the state-space flows cannot be obtained by integrating the force terms (the reaction terms of the ODE system). This issue can be bypassed by introducing stochasticity into the system. In a stochastic system, each state **x** (here $\mathbf{x} = (x, y) = (u_L, u_R)$) is described using a probability in time and state space position **x**, P(**x**, t). The time evolution of the P(**x**, t) not only depends on the forces that drive the system, but also the stochastic transitions between adjacent points in the state space. This can be formalized using a Fokker-Planck equation that captures the interplay between deterministic and stochastic nature of the system and is given by,

$$\frac{\partial P(u_L, u_R, t)}{\partial t} = -\frac{\partial(G_1 P)}{u_L} - \frac{\partial(G_3 P)}{u_R} + D\left(\frac{\partial^2}{\partial u_L^2} + \frac{\partial^2}{\partial u_R^2}\right)P \tag{21}$$

where $D$ is the diffusion constant associated with stochastic transitions, $G_1$ and $G_3$—as in Eq (3). By numerically solving Eq (21), the asymptotic state of the probability distribution, $P_{ss}$, given by the limit P(**x**, $t \to \infty$), is estimated. Analogous to the equilibrium state, an approximate expression for the quasi-potential is then given by, $Q(\mathbf{x}) \approx -ln(P_{ss})$. The Fokker-Planck equations were solved numerically using the *Python* package provided in [38], with $D = 0.02$. The two dimensional grid on which the system is solved has a spatial step size 0.02.

To quantify the landscapes, the Gaussian curvature $K$ of the landscapes given by:

$$K(x, y) = \frac{Q_{xx}Q_{yy} - Q_{xy}^2}{\left(1 + Q_x^2 + Q_y^2\right)^2} \tag{22}$$

is used, where $Q_x = \frac{\partial Q}{\partial x}$, $Q_{xx} = \frac{\partial Q_x}{\partial x}$, $Q_y = \frac{\partial Q}{\partial yy}$, $Q_y = \frac{\partial Q_y}{\partial y}$, $Q_{xy} = \frac{\partial Q_x}{\partial y}$ are the first and second order partial derivatives of the quasi-potential surface. State space regions with positive $K$ values were identified using a threshold given by $K_{mean} + 0.1K_{std}$. The boundary that determines the asymptotic behavior of the trajectory in the vicinity of a steady state, $Q_{bound}$, is estimated as the mean of the quasi-potential values at the boundary of the identified region that satisfies the condition that the slopes are distributed around zero (Fig 1F).

## Model implementation

The models were implemented using a custom-made *Python* code. The PDE solving method that we have used is as follows. Given a generic reaction diffusion system on a 1D domain (equivalent to Eq (1), where $\theta \in [\theta_{min}, \theta_{max}]$, the domain is first discretized to N = 20 spatial

bins with uniform bin size $\delta\theta = \theta_{i+1} - \theta_i$ for $i = 1, 2, \ldots, N-1$. The discretized version of the PDE then becomes

$$\frac{\partial u_i}{\partial t} = f_u(u_i, v_i) + D_u \frac{\partial^2 u_i}{\partial^2 \theta}$$

$$\frac{\partial v_i}{\partial t} = f_v(u_i, v_i) + D_v \frac{\partial^2 v_i}{\partial^2 \theta}$$

(23)

where $u_i = u(\theta_i, t)$, $v_i = v(\theta_i, t)$. Conversion of this PDE to ODE is then done using the method of lines [39] where the second order partial derivative terms are approximated using finite difference method. This enables us to rewrite the equations with partial derivatives in t as total derivatives,

$$\frac{du_i}{dt} = f_u(u_i, v_i) + \frac{D_u}{\delta\theta^2}(u_{i+1} - 2u_i + u_{i-1}) + O(\delta\theta^2)$$

$$\frac{dv_i}{dt} = f_v(u_i, v_i) + \frac{D_v}{\delta\theta^2}(v_{i+1} - 2v_i + v_{i-1}) + O(\delta\theta^2)$$

(24)

Depending on the type of boundary conditions, equations at the boundary bins $i = 1$ and $i = N$ are fixed. For example, for periodic boundary conditions, two fictitious bins $\theta_{-1}$ and $\theta_{N+1}$ with constrains $\theta_{-1} = \theta_N$ and $\theta_{N+1} = \theta_1$ are considered, which allows to re-write the equations at the boundary as:

$$\frac{du_1}{dt} = f_u(u_1, v_1) + \frac{D_u}{\delta\theta^2}(u_2 - 2u_1 + u_N) + O(\delta\theta^2)$$

$$\frac{dv_1}{dt} = f_v(u_1, v_1) + \frac{D_v}{\delta\theta^2}(v_2 - 2v_1 + v_N) + O(\delta\theta^2)$$

(25)

and

$$\frac{du_N}{dt} = f_u(u_N, v_N) + \frac{D_u}{\delta\theta^2}(u_{N+1} - 2u_N + u_1) + O(\delta\theta^2)$$

$$\frac{dv_N}{dt} = f_v(u_N, v_N) + \frac{D_v}{\delta\theta^2}(v_{N+1} - 2v_N + v_1) + O(\delta\theta^2)$$

(26)

This set of equations can now be solved using any standard numerical solver for ODEs. In order to ensure numerical stability of the solutions, we have used explicit Runge-Kutta method of order 5(4) with adaptive time step $dt$ (implemented using *solve_ivp* package in *Python*). Truncation error of the order $O(dt^6)$ was sufficient to capture sharp transitions.

The external perturbations into the system of ODEs is modeled as a Wiener process where Gaussian white noise is introduced as an additive term at each time step. This results in a stochastic differential equation (SDE) in the Ito form

$$du_i = [f_u(u_i, v_i) + \frac{D_u}{\delta\theta^2}(u_{i+1} - 2u_i + u_{i-1})]dt + \sigma dW(0, 1)$$

$$dv_i = [f_v(u_i, v_i) + \frac{D_v}{\delta\theta^2}(v_{i+1} - 2v_i + v_{i-1})]dt + \sigma dW(0, 1)$$

(27)

where $dW(0, 1)$ is the Gaussian white noise term with unit variance and $\sigma$ is the noise intensity. Euler-Maruyama algorithm (implemented using *sdeint* package in *Python*) was then used to solve this system. For the RD simulations, the stimulus gradient was generated using Gaussian

function from *scipy.signal.windows* in *Python*. This package truncates the Gaussian function which otherwise extends from $-\infty$ to $+\infty$ within a given window. For a window of length $N$ (in our case $N = 20$), Gaussian profile is constructed using the expression $s(n) = s_0 e^{\frac{-1}{2}\left(\frac{n}{w}\right)^2}$ where $n \in \left[-\frac{N-1}{2} : \frac{N-1}{2}\right]$, $s_0$ is the signal amplitude, and the variance is $w = \frac{N-1}{2\alpha}$. Varying the value of the constant $\alpha$ results in Gaussian profile of varying spread. The generated Gaussian will be maximum at the center and when overlayed on the membrane (S1(A) Fig top, bottom) results in a maximum at $\theta = \pi$ with negligible discontinuity at $\theta = 0, 2\pi$. For Figs 1C and 3 (except differently specified), $s_0 = 0.02$ and $\alpha = 2$ is fixed to have 25% of width at half maximum. For the RD simulations in Fig 3A, $s_0$ is systematically varying while keeping $\alpha$ fixed, whereas for S3(D) Fig, $\alpha$ is systematically varied. For simulation of the one dimensional projection models, a step like signal function was used (S1(B) Fig), with signal amplitude $s_L$ and $s_R$ (generally set to 0).

## Model comparison

In order to compare polarization features arising from the different types of dynamical mechanisms, we quantified several metrics: polarization ratio $\left(\frac{u_{\theta=\pi}}{u_{\theta=0}}\right)$ to steep and shallow gradients quantified via a stimulus difference ($sd = (s_{\theta=\pi} - s_{\theta=0}) \times 100$), time to reach stable polarization at a threshold signal amplitude inducing polarization ($sd_{thresh}$), polarization ratio to signals with increasing offset, time necessary for polarization reversal/resolving simultaneous stimuli and subsequent polarization amplification, and response to consecutive stimuli (using signal integration index). In order to estimate the polarization time (Fig 3A), $u(\theta, t)$ was normalized between max and min values to enable model comparison. Polarization time was then estimated as the first time point at which the normalized response reaches within a small window ($\pm 10^{-2}$) around the mean of the last 100 time points during gradient stimulation. The threshold for activation ($sd_{thesh}$) represents the minimal stimulation amplitude for which stable polarization was achieved, and was estimated from Fig 3A as the $sd$ where 50% of the maximum polarization ratio is reached. $sd_{thresh}$ was manually set for the LEGI model to 0.5%, and for the Turing-model to 0.1%, as both systems exhibit spurious activation to noise. For model responsiveness to signals with an offset, the maximal signal amplitude was systematically varied by adding an increasing off-set amplitude to the $sd_{thresh}$. Polarization reversal and resolving times were estimated equivalently to the polarization time. For the reversed polarization in Fig 3D, the Turing and the Wave-pinning models are not depicted, as they did not re-polarize in the time frame of 1000s. The signal integration index in Fig 3F is estimated as $\frac{([u_{\theta=\pi,a}]-[s_{\theta=\pi}])}{t_{max}}$ where $[u_{\theta=\pi,a}]$ is the total duration in which $u_{\theta=\pi} > 0.5$, $[s_{\theta=\pi}]$ is the total duration of signal gradient stimulation, and $t_{max}$ is the total simulation time. The spurious activation in the absence of signal in S3(A) Fig is performed by considering a random perturbation around the homogenous steady state $(u_s, v_s)$ which is implemented as $(u_s + \xi_{per}r, v_s - \xi_{per}r)$ where $r$ is a random number between [0, 1]. For the LEGI model, the perturbation is also implemented on the $w_s$ variable.

## Supporting information

**S1 Fig. Characterizing non-autonomous state transitions for the model in Fig 1A.** Schematic representation of the gradient signal implementation in (A) the RD simulations, corresponding to Eq (1), and (B) the one-dimensional projection model, corresponding to Eq (3). (C) Time series of $u$ and the quasi-potential landscape transitions during a transient step-like stimulation for organization in region I (top), III (middle) and IV (bottom) corresponding to Fig 1B (same equations and parameters). Green shaded region: signal interval. Circle/square/

triangle: non-polar/polar/transient-polar (memory) state. (D) Exemplary estimate of Gaussian curvature (middle) and corresponding slopes distribution in $(x, y) = (u_L, u_R)$ direction for each of the identified regions. Slopes distribution around 0 in both direction in conjunction with positive curvature uniquely determines a well (stable steady state) in the potential landscape. Description as in Fig 1F.
(TIF)

**S2 Fig. Polarization response of the LEGI, Turing-like and Wave-pinning models.** (A) Spatial-temporal response (kymographs) of the membrane-bound active component of the three models. Parameters as in Fig 2, except for $D_u = D_w = 0.5 \mu m^2 s^{-1}$, $D_v = 10 \mu m^2 s^{-1}$ for the LEGI, and $D_u = 0.1 \mu m^2/sec$, $D_v = 10 \mu m^2 s^{-1}$ for the Turing and Wave-pinning models. (B) Temporal $u$ profile for the LEGI model, corresponding to Fig 2D and 2E.
(TIF)

**S3 Fig. Spatial-temporal responses of the four different polarity models.** (A) Quantification of spurious activation for increasing perturbation amplitude around the homogeneous steady state. Colors as in Fig 3A. (B) Schematic representation of gradient stimulation with an offset along the cell membrane contour. (C) Kymographs depicting the spatial-temporal response of each of the models to reversal of gradient stimuli (black horizontal line). Red horizontal line: time point when stable reversed polarity is established. (D) Comparison of repolarization in the Wave-pinning (left) and the SubPB (right) models, for varying stimulus width and maximal stimulus amplitude. (E) Kymographs depicting the spatial-temporal response of each of the models stimulated with simultaneous signals with different amplitudes from opposite cell ends. Red horizontal line: time point where stable polarization with unique axis was established. (F) Exemplary temporal response to consecutive signals from same direction (left: LEGI, Wave-pinning and SubPB; right: Turing model).
(TIF)

## Author Contributions

**Conceptualization:** Akhilesh Nandan, Aneta Koseska.

**Formal analysis:** Akhilesh Nandan.

**Funding acquisition:** Aneta Koseska.

**Investigation:** Akhilesh Nandan, Aneta Koseska.

**Software:** Akhilesh Nandan.

**Supervision:** Aneta Koseska.

**Validation:** Akhilesh Nandan.

**Visualization:** Akhilesh Nandan.

**Writing – original draft:** Akhilesh Nandan, Aneta Koseska.

**Writing – review & editing:** Akhilesh Nandan, Aneta Koseska.

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
