## [Decision Letter · Decision Letter 0]

29 Apr 2023

Dear Dr Koseska,

Thank you very much for submitting your manuscript "Non-asymptotic transients away from steady states determine cellular responsiveness to dynamic spatial-temporal signals" for consideration at PLOS Computational Biology.

As with all papers reviewed by the journal, your manuscript was reviewed by members of the editorial board and by several independent reviewers. In light of the reviews (below this email), we would like to invite the resubmission of a significantly-revised version that takes into account the reviewers' comments.

Especially focus on clarifying all details of the model and ensure all results are reproducible and necessary codes are deposited.

We cannot make any decision about publication until we have seen the revised manuscript and your response to the reviewers' comments. Your revised manuscript is also likely to be sent to reviewers for further evaluation.

Sincerely,

Attila Csikász-Nagy

Academic Editor

PLOS Computational Biology

Douglas Lauffenburger

Section Editor

PLOS Computational Biology

Reviewer's Responses to Questions

**Comments to the Authors:**

Reviewer #1: This manuscript concerns generic models of cell polarization in response to imposed spatially non-uniform cues. In particular, it addresses the question of what type of cellular response mechanism can respond to physiological cues in a robust way (definitively and not too sensitively to weak/stochastic cues) while maintaining the ability to adapt to new directional cues.

The authors focus on three broad classes of model that have been proposed in previous publications: local excitation global inhibition (LEGI), Turing-type, and wave-pinning. Using a combination of bifurcation and local stability analysis, and explicit consideration of transient responses to time-varying spatial cues, they illustrate the potential importance of slow transient dynamics in cellular responses. By mapping out the phase space dynamics, they show that LEGI and Turing-type models are not able to account for adaptability, and that the parameter regime typically studied in wave-pinning models also does not account for this. Building on a proposal the authors made for a specific polarization model in a previous publication (Nandan et al., 2022), the authors demonstrate that a different parameter regime for a generic wave-pinning model exhibits criticality close to a saddle-node bifurcation that is associated with slow transient responses to changes in spatial cue. These can result in cellular memory that can underlie a combination of persistence in the face of fluctuating cues with the ability to reorient to new spatial cues.

The general approach adopted by the authors is an important and valuable one, and the focus on transient dynamics in explicitly non-autonomous systems (i.e. those with explicitly time-varying inputs) is quite novel in this context. I agree with the authors that transients and non-autonomy are a neglected aspect of models of cellular dynamics. One reason for this is the challenge of saying anything concrete about possible behaviours in a way that goes beyond simple numerical simulation. An appealing feature of the work reported here is the use of a combination of bifurcation analysis, the construction of pseudo-potential landscapes, and local perturbation analysis to analyse model responses to time-varying cues, focusing on transient behaviours.

So, I am supportive of the approach used here, and of the message. I think it is an important contribution to understanding the mechanisms of cell polarization and to the necessary move towards exploring transient dynamics of systems in response to time-varying stimuli. However, I think the manuscript needs quite a lot of work to get it in to an acceptable form.

The major problem is that the models are not specified properly and/or clearly -- their functional form and the parameters used. Specifically, there are two principal forms of the models in the manuscript: the spatially continuous form on a circle and the spatially discrete form in two compartments (left and right). It is almost never clear which model is being used. For example, Fig 1C seems to show a simulation of the continuous model (with a Gaussian signal that does not quite correspond to the one stated in Materials and Methods, and with no “spread” or “amplitude” parameters specified), but Fig 1E has an “s_{left}”, which presumably refers to some discrete signal in the discrete model (I honestly don’t know; as far as I can tell, the meaning of “s_{left}” is never given.)

Specific points:

Eq. (1): Is the way that the signal is introduced (additive on the u equation only) correct? If so, it breaks mass conservation (production of u without any effect on v). Should it also be subtracted from the v equation?

Line 60: Perhaps a little pedantic, but f_u and f_v: [0, 2 pi) x R � R. Unless you want to put a modularity condition on the first R.

Line 63: v is defined as a cytosolic component (see also Fig. 1A). However, it is treated in the model as a membrane component (it diffuses in the membrane.) If it was truly cytosolic, it would diffuse “across” the cell, giving a non-local connection in theta. I presume (without reminding myself) that this is the form used (and justified) by Mori et al., but I think it needs a bit of clarification here if v is going to be described as cytosolic.

Line 64: I think mass conservation should be an integral condition (u + v integrated over theta), not just u+v = c_{total} (which doesn’t make sense if u and v are dependent on theta.) See above: the current form of (1) breaks mass conservation unless s = 0.

Fig. 1 legend: 1B introduces regions I-IV. However, these are a bit “orphan” at this stage, as they’re not referred to. I think lines 72-106 could be rewritten to make the overall scheme clearer. I suggest that it would be helpful at this stage to explain what the four regions of the wave-pinning model correspond to. As c_{total} increases we have:

I: monostable – only non-polarized HSS exists and is stable;

II: critical – only non-polarizes HSS exists, but there is a ghost of a saddle-node bifurcation;

III: tristable – stable HSS coexists with two stable polarized states (L and R polarized);

IV: bistable – HSS is unstable; two stable polarized states (L and R).

It is then easier to explain the point about criticality: In regions III and IV, a transient signal can/will irreversibly shift the system to one of the stable polarized states; in region II, a transient signal will move the system to a polarised state from which it takes a long time to return to the stable HSS (memory, due to the ghost).

I appreciate that this is in there, but I don’t think it is made very clear (and you’re not really using the I—IV notation in the text at this stage.) I think you only finally start to make the point clearly in lines 177-193; the points contained here could usefully come earlier.

Fig 1D: “response” is not really defined at this point.

Line 85: “critical organization” is not well defined here. See above for a suggestion.

Lines 95-96: It’s a matter of semantics, but I think “a step-like approximation approach to the change in the parameter values” is potentially a bit misleading. I would be inclined to describe s(theta, t) as an imposed signal rather than a parameter.

Line 96: “discreetly” should be “discretely” (unless you’re trying to slip it past the reader without them noticing.)

Lines 107-119: I really don’t like the way this little interlude to point out the difference of the critical region to a bistable model disrupts the flow of the analysis of the wave pinning model. I understand the point that the authors want to make, and why they want to make it, but I think it is out of place. If it has to be in, I think it would be much better after line 158. However, I think it would be much more appropriate to use the bistable region (region IV) of the current wave pinning model to make the comparison, rather than a seemingly ad hoc bistable model (non-mass conservative?) I’d be much happier to see a comparison of transient responses / succession of steady states in the four regions of the main model studied here, rather than the current comparison to a totally different model brought in solely for the purposes of making a point about non-transient responses.

Lines 165-167: Please refer to the equations for the LEGI and Turing models here (Eqs. (18) and (19), though the latter need a bit of work, I think -- see below.)

Line 198: I think s_{left} and s_{right} have still not been defined at this point… And, unless I am missing something, (s_{left} - s_{right}) x 100 is not a percentage difference. For that, you’re going to need a denominator something like 0.5x(s_{left} + s_{right}).

Lines 203-204: “the polarization ratio achieved by the LEGI-type model is relatively small (<1)”. Something doesn’t seem right here. The polarization ratio is defined as u_{left}/u_{right}, which must be >1 (unless the system has polarized in a direction opposite to the applied signal.) There seems to be a problem with pd and % in this paragraph.

Lines 208, 211-212: the notion of “spurious activation” is raised, but is not defined. If it is being used as a criterion for judging the appropriateness of different models, a bit more work needs to be done to say what it is.

Line 216 et seq.: It is not clear to me what a “gradient with off-set” is.

Line 223: “where as” should be “whereas”.

Line 310: The characteristic equation relates to spatially uniform perturbations, and does not result from substituting (4) in (3) as claimed (which would have diffusion terms.)

Lines 314-315: the sentence doesn’t have an ending.

Eq. (7): It would be better to put the variables in in the RHS (i.e. G_1(u_{left},…)).

Also, this may be personal preference, but I think it would all be a lot more readable if u_{left} was replaced with u_L, etc. The “= G_1” etc. should be something like “:= G_1” to make it clear that this is being taken as a definition of G_1 (rather than an unspecified identification of two things, one of which is not defined.)

Line 338: The G_i are defined as “reaction terms”, but they encode both reaction and diffusion in the RD equations. Conventionally, “reaction term” is used only for the diffusion-independent part. It also feels a bit late after Eq. (7) to say what the G_i are…

Lines 340-341: I don’t understand what the authors mean by evaluating the determinant of the Jacobian for the odd mode of perturbation. The Jacobian is evaluated at U_S; it does not depend on the mode of perturbation (it is a local property of U_S.)

Eq (9): It is not clear what F_(lambda) is; it is not defined (I think this is a typo, or I’m missing something.) Also the footwork to get from (8) to (9) is a bit opaque.

Lines 330-342 (Example): I can’t see what role this tangent into an example is playing, and it disrupts the flow of the general analysis. I find it really strange how it ends abruptly with Eq. (9), which doesn’t appear to then go anywhere (or tell us anything in particular.) What is Eq. (9) doing in the narrative? If it were up to me, I would cut this tangent completely, or defer it until after the completion of the general analysis (Eq. (15)) and revise it so it actually adds something.

Line 349: It should be f_u(u,v) = (k_0 + gamma u^2 /K^2)v – delta u. There is a K^2 missing.

Lines 360-369: It is stated that “Given that D_v >> D_u…”. Why is this “given”? Is it given for all models studied? Is it an assumption (required to do LPA), or must it be true for all models of polarization? This section is a bit too vague to be of much help to a reader who hasn’t read Holmes et al. It would be better either to just leave a reference to Holmes et al / Grieneisen or to expand it so that it actually gives more useful information for the reader. Either would be appropriate, but as it stands it doesn’t really help.

Lines 373-374: The values for c_{total} for wave-pinning and SubPB are very close (2.25 vs 2.21 – less than 2% difference.) Does it really make sense to claim that cells could accurately “position themselves” so accurately (to take advantage of the saddle node ghost in the way described in this study?) What are the values for D_u and D_v?

Lines 376-379: The Turing model is not well defined. What is f_v? Is it just -f_u (mass conservation)? If so, say so. What are the values for D_u and D_v?

Lines 390-392: A few questions: 1. Is it right to add the signal term s only to the u equation?; 2. I think the last term in the expression for f_u should be b_1 u v , not g_1 u v; 3. Unlike the other models, this one appears to not be mass-conserving (unless I’m missing something.) Is that right? If so, it would seem to make an even stronger case for using the bistable region of the wave pinning model (region IV) as a comparator rather than this seemingly ad-hoc bistable model; 4. The signal is not properly defined. It just says what the maximum amplitude of signal is without specifying its profile in space and time.)

Lines 415-422:

1. “One dimensional domain of length L = 2 pi R with R = 2” doesn’t seem to fit with the rest of the manuscript. For a start, all parameters are dimensional, so R would have to be. But more importantly, the models are posed on the circle, with angular variable theta in [0, 2 pi). This needs sorting out;

2. This section states, almost as an aside, that the ODEs were converted to SDEs with “noise intensity” sigma. A bit more detail is needed here (I’m meant to be able to reproduce the results, and there is no unique way of going from an ODE to an SDE.);

3. “initial conditions are chosen to be around the homogeneous steady steady state” is a bit too vague. The intial conditions need to be stated more concretely than “around”;

4. “The stimulus gradient was implemented using a Gaussian function of the form s(theta) = ...”. Two points here: first, sigma is used again, when it has just been used for noise intensity in the SDEs. Use two different symbols. More importantly, it is not clear that this form of s(theta) is actually being used in the simulations. For a start, it seems signal is centred on theta = pi, to give a two-sided Gaussian; the form stated would be a one-sided Gaussian with a discontinuity at theta = 0 / 2pi. But actually I can’t find any instance where I am certain a Gaussian signal is being used in simulations – it seems to always be a piecewise constant signal (most commonly s_{left} and s_{right}.) This is all very confusing. It is necessary to be explicit and clear about what signal is being applied, for what time duration (beyond just putting green lines on figures) in each numerical simulation. As it is, the reader is left guessing / trying to infer what signal was applied from the results… If the Gaussian form is used in simulations, I couldn’t spot any indication of what value of sigma (in this sense) was used.

Some problems with the references:

The reference “Veronica, G. (2009)…” should be “Grieneisen, V. (2009)…”

Jilkine, A., & Edelstein-Keshet, L. (2011). A comparison of mathematical models for polarization of single eukaryotic cells in response to guided cues. PLoS computational biology, 7, e1001121.

Levine, H., Kessler, D. A., & Rappel, W. J. (2006). Directional sensing in eukaryotic chemotaxis: a balanced inactivation model. Proceedings of the National Academy of Sciences, 103, 9761-9766.

Otsuji, M., Ishihara, S., Co, C., Kaibuchi, K., Mochizuki, A., & Kuroda, S. (2007). A mass conserved reaction–diffusion system captures properties of cell polarity. PLoS computational biology, 3, e108.

Verd, B., Crombach, A., & Jaeger, J. (2014). Classification of transient behaviours in a time-dependent toggle switch model. BMC systems biology, 8, 1-19.

A plea: Thank you for including line numbers, but the small font size and single spacing made this manuscript really, really hard to read carefully. Please consider submitting with at least 1.5 line spacing in future. It is specified as a submission requirement for a reason.

Reviewer #2: I found the text very difficult to read, and perhaps because of that I misunderstood what is it that the authors really did. If I understood correctly, they are exploring the well-known wave-pinning model in the presence of the signal that changes in time. They both analytically and numerically investigate the transient polarized cell states in this model discuss important concepts of memory and responsiveness of the cell. I sense that the paper is valuable, however 3 major revisions have to be made:

1) drastically rewrite and reorganize the manuscript to make the main points more concisely and clearly - in the present form maybe only ~ 10 readers will be able to appreciate the study.

2) reduce the drama in the intro and discussion - the point that existent model cannot exhibit the dynamics that the authors found is moot: the authors are using exactly the models they are criticising, just with the additional signal term. By the way, there were numerous numerical studies that used the same models with dynamic signal.

3) Explain clearly what is the novelty compared to your own eLife paper - it seems to me that all the main points were already made there.

Reviewer #3: Re: Review of Non-asymptotic transients away from steady states determine cellular responsiveness to dynamic spatial-temporal signals

The paper argues for a paradigm shift when considering models for cellular polarity. Cell polarization is crucial for cells to establish a front and back, and to organize cell cytoskeletal processes to facilitate cell migration. Understanding the biochemical networks establishing polarity is crucial to understanding cell migration. Polarization models most commonly are formulated as reaction-diffusion equations. Two main classes exist: (1) Turing based mechanisms; and (2) wave-pinning mechanisms.

The authors of this study argue that none of the existing models can recapitulate both (1) maintenance of a polarized state longer than the signal duration, and (2) retain responsiveness to novel signals. The authors encourage modellers in the field to embrace non-autonomous and non-asymptotic signalling dynamics as a solution. In particular, the authors demonstrate their argument with their recently developed SubPB dynamics. While I don't fully agree with the authors assessment in this regards (see some references suggesting otherwise below). I find that the authors point that we should consider non-autonomous dynamics an important one, and the referenced papers do essentially consider changes in time.

In my opinion the paper addresses an important question i.e.\\ how to develop signalling networks that have all features observed in experiments. Further, in my opinion the paper argues for an important often neglected piece, that these signalling networks must respond to spatio-temporal variations. Frequently this means that transients and non-asymptotic dynamics are crucial for us to understand signalling networks.

The paper is generally well written, albeit confusing at times (see specific comments below). The major challenge for me was to understand what the SubPB model is, how it is defined, how it is different to Turing and wave-pinning models. Having worked on these models before, I was excited to understand how the authors addressed the above mentioned challenges, and tried for significant time to understand how the SubPB model works. However, I did not succeed. This is amplified by Figures that don't fully explain what is shown i.e. (conditions to replicate the figures, and sometimes what equations are used). Given this, I currently don't think that this paper is reproducible (code is not available, as far as I can tell). Having said this, I believe that these things can be fixed, and would encourage the authors to carefully revise their manuscript, with readability and reproducibility in mind. In particular, I would encourage the authors to not refer any of the mathematical and modelling details to their recent eLife publication.

Once the paper is revised, I will support its publication in PLOS CB.

Major Comments

1) The major issue I encountered while reading this paper is that I am not clear what the SubPB mechanism is, and more importantly how it is different to the wave-pinning model. I did look at the referenced eLife article, but that didn't help. In my opinion, it would greatly improve the readability of this paper if the authors clearly define what the SubPB model is. Clarifying this may also address point (2) below.

2) Equation (1). When s(t, theta) = 0, then given the authors choice of nonlinearities this becomes the wave-pinning model as introduced by Mori et al, featuring mass-conservation and stalling waves. The way the external perturbation is added here destroys the mass-conservation property. Other studies [3] and [4] below consider perturbations that leave the mass conservation property in tact, while [5] considered relaxing this assumption. Based on reference [5], I have the impression that relaxing the mass conservation property, substantially changes the dynamics of these models.

Having said all of this, I am unclear whether it's the authors intention to break mass conservation or not, and whether breaking mass conservation leads to different model behaviours. Once again, I think that I'm mainly unclear about the authors' assumptions, and what exactly the SubPB model is.

3) Almost all of the Figures appear to be lacking the required information to reproduce the images i.e.\\ numerical methods used, parameters, and initial conditions for the shown simulations.

4) The matched asymptotic analysis in Mori et al (2011) shows that the transition point from u_left to u_right is a function of the total mass in the system. A similar projection (as in this paper) was employed by Walther et al (2012). In this work, the transition point was still a function of the total mass, in reference to the asymptotic analysis by Mori et al (2011). Here the authors employ a projection that seemingly has a fixed transition point. How do the authors justify this?

Even in Turing systems without mass conservation the transition point in a ``mesa-pattern'' is not centered in the domain. In fact matched asymptotic analysis by McKay and Kolokolnikov (2013) shows that its a function of the nonlinearities used.

Given this how good of an approximation is the used projection?

Studies looking at similar questions:

There are a number of studies both computational and theoretical that have examined similar questions recently, and not so recently, that in my opinion would merit mentioning.

1) Polarity and mixed-mode oscillations may underlie different patterns of cellular migration; Plazen et al (2023) in Scientific Reports, appears to address a similar question. In particular, the authors were motivated to modify and extend existing polarity models by experimental data showing cell polarization reorientation.

2) Deterministic Versus Stochastic Cell Polarisation Through Wave-Pinning (2012). A study that uses a similar projection, as considered in the present paper.

3) Cell Repolarization: A Bifurcation Study of Spatio-Temporal Perturbations of Polar Cells (2022) BMB. A study which appears to demonstrate that the wave-pinning models to retain sensitivity to subsequent signals, and re-orientation of the polarized states is possible. While this study is autonomous, it seems it could be re-interpreted in a non-autonomous setting.

4) Dynamics of localized unimodal patterns in reaction-diffusion systems for cell polarization by extracellular signaling (2018) SIADS. This paper considers the same setup as this study. A wave-pinning model on a periodic domain, with signalling inputs. This paper also suggests that existing wave-pinning models can respond to subsequent signals.

5) A Model for Cell Polarization Without Mass Conservation (2017) SIADS. Considers cell polarization models without mass-conservation. These models appear to be quite different in terms of the featured dynamics.

Minor Comments

- In Figure 1A: The projected system appears to have non-periodic boundary conditions. This is of course fine, but the authors should mention why that is, and how to map the periodic to non-periodic domains. This is something that confuses many.

- Figure 1F/G: Which model is this plot for? The projection or the full RDE?

- Supp Figure 3: Why is the SubPB mechanism not shown here?

- Line 81: Which one-dimensional model are you referring to? The projection or the PDE?

- Line 239: What characteristics?

- Line 315: The sentence seems incomplete.

- Line 330: Again I don't understand what the difference between wave-pinning and SubPB is. Later on Line 373 the authors state a c_total for the SubPB model, but I don't think it conserves mass. Second the wave-pinning models are only described by equation (1) if s = 0.

- Line 334: The vectors should not be inline the text. Maybe use the transpose to make this look better.

- Line 416: For the simulations the authors use only 20 discretization points. That seems really small. That gives a spatial step size of larger than 1/2, and since the global error of a central difference scales like O(h^2) a global error of larger than a 1/4. For a RDE with sharp transitions, I would expect that such a coarse domain discretization would not be able to accurately describe the evolution of the sharp interface.

- Line 503: The last and first name of the reference are flipped.

**Have the authors made all data and (if applicable) computational code underlying the findings in their manuscript fully available?**

Reviewer #1: **No: **I can't see any indication in the manuscript that code is available (e.g. on github).

Reviewer #2: Yes

Reviewer #3: **No: **I may have missed it, but as far as I can tell there is no link to a repository with the used code.

PLOS authors have the option to publish the peer review history of their article (what does this mean?). If published, this will include your full peer review and any attached files.

Reviewer #1: No

Reviewer #2: No

Reviewer #3: No
---

## [Decision Letter · Decision Letter 1]

25 Jul 2023

Dear Dr Koseska,

We are pleased to inform you that your manuscript 'Non-asymptotic transients away from steady states determine cellular responsiveness to dynamic spatial-temporal signals' has been provisionally accepted for publication in PLOS Computational Biology.

Best regards,

Attila Csikász-Nagy

Academic Editor

PLOS Computational Biology

Douglas Lauffenburger

%CORR_ED_EDITOR_ROLE%

PLOS Computational Biology

Reviewer's Responses to Questions

**Comments to the Authors:**

Reviewer #1: I remain supportive of the work, and the authors have made a significant effort to address my concerns about the clarity of the descriptions of the models and numerical simulations. I therefore recommend publication.

Reviewer #2: I am satisfied with the revisions

**Have the authors made all data and (if applicable) computational code underlying the findings in their manuscript fully available?**

Reviewer #1: Yes

Reviewer #2: Yes

PLOS authors have the option to publish the peer review history of their article (what does this mean?). If published, this will include your full peer review and any attached files.

Reviewer #1: No

Reviewer #2: No

---

## [Editor Report · Acceptance letter]

5 Aug 2023

PCOMPBIOL-D-23-00259R1 

Non-asymptotic transients away from steady states determine cellular responsiveness to dynamic spatial-temporal signals

Dear Dr Koseska,

I am pleased to inform you that your manuscript has been formally accepted for publication in PLOS Computational Biology. Your manuscript is now with our production department and you will be notified of the publication date in due course.

With kind regards,

Zsofia Freund
